# Dielectric Surface Flashover under Long-Term Repetitive Microsecond Pulses in Compressed Gas Environment

**DOI:** 10.3390/ma14123343

**Published:** 2021-06-17

**Authors:** Tianyu Lin, Yongpeng Zhang, Zhijian Lu, Zhengwen Wang, Peng Wei, Chengying Liu, Lanjun Yang

**Affiliations:** State Key Laboratory of Electrical Insulation and Power Equipment, School of Electrical Engineering, Xi’an Jiaotong University, Xi’an 710049, China; lty091513@stu.xjtu.edu.cn (T.L.); zhangyp302@stu.xjtu.edu.cn (Y.Z.); luzhijian0805@xjtu.edu.cn (Z.L.); wzw0627@stu.xjtu.edu.cn (Z.W.); weipengroc@stu.xjtu.edu.cn (P.W.); struggling@stu.xjtu.edu.cn (C.L.)

**Keywords:** stage insulator, flashover characteristics, microsecond repetitive pulses, flashover path, accumulation effect

## Abstract

As a key component of a high-power microwave (HPM) system, a multi-gap gas switch (MGS) has recently developed insulation failure due to surface flashover. Although design criteria for surface insulation have been put forward, it is still not clear how the insulation in this case deteriorated under long-term repetitive microsecond pulses (RMPs). In this paper, flashover experiments under RMPs were carried out on various dielectric surfaces between parallel-plane electrodes in SF_6_ and air atmospheres, respectively. Based on tests of the surface insulation lifetime (SIL), an empirical formula for SIL prediction is proposed with variations of insulator work coefficient *λ*, which is a more suitable parameter to characterize SIL under RMPs. Due of the accumulation effect, the relationship between *E*/*p* and *pt*_delay_ varies with the pulse repetitive frequency (PRF) and SIL recovery capability decreases with an increase in PRF and surface deterioration is exacerbated during successive flashovers. It is concluded that the flashover path plays a crucial role in surface insulation performance under RMPs due to the photoemission induced by ultraviolet (UV) radiation, signifying the necessity of reducing surface paths in future designs as well as the improvement of surface insulation.

## 1. Introduction

A self-break repetitive MGS [1], shown in Figure 1, composed of a one-trigger stage and five rimfire stages in series, was developed to meet the demands of operating at a higher voltage and a higher PRF for the HPM generator as well as having a more compact structure. Rimfire stages in series were utilized to improve the breakdown voltage and the recovery capability after suffering repetitive pulses compared with the single-stage switch. Stage insulators are evenly placed between the rimfire stage electrodes for insulation and mechanical support. The operating voltage of the switch can be adjusted by varying the insulator thickness. Meanwhile, the insulation capability of the stage insulators contributes to the performance of the switch. According to the Martin formula [2], a longer lifetime demands insulators with a higher hold-off field strength. In our investigations, the working conditions of the MGS in the HPM generator are less than 780 kV (one rimfire stage is 130 kV) with a rise time of ~30 µs at 50 Hz in a 0.4 MPa SF_6_ environment.

Recently, a failure event in the MGS developed that resulted from the degradation of stage insulators after only a few shots and obvious spark tracking along the insulator surface was observed. Conventionally, for a dielectric/gas composite insulation system between a couple of parallel-plane electrodes, insulation failures are mostly caused by surface flashover instead of bulk breakdown, since the interface is always the weakest part under RMPs. In previous investigations, factors affecting surface flashover, mainly in gas-insulated switchgears (GISs), were systematically studied [3,4,5] and various methods have been developed to improve bulk and surface insulation strengths under AC and DC conditions [6,7,8]. When it comes to pulsed flashovers, the physical processes and mechanisms of solid-vacuum surface flashovers have been discussed in detail in pulsed power systems [9,10]. Andreas’ previous research [11,12] mainly focused on the effects of gas species, humidity, roughness, and UV radiation on pulsed dielectric surface flashovers under atmospheric conditions, but little is mentioned regarding repetitive pulses excitations [13]. In addition, the electric field distributions of test electrodes commonly used in previous investigations, such as needle-plate electrodes [14] and finger-type electrodes [9,10], are not applicable for the working conditions of stage insulators in MGSs.

In fact, stage insulators suffer from stronger electric fields than expected because the breakdowns of the rimfire stages in series are triggered by over-voltage from closure of the trigger stage. Even after spark discharge channels between the stages have formed, it can be assumed that the transient nature of the switching process is attributed to the sufficiently strong electric field that remains along the surface of the stage insulators [11]. Furthermore, the insulators are exposed to optical radiation, especially UV content, which is gathered from the spark discharge channels when gap breakdown occurs. Based on the continuous working mode of HPM generators, it should be noted that the effects of PRF on lowing surface insulation capabilities must be considered; however, there are few electric parameters mentioned in previous investigations other than flashover voltage and time delay to describe surface insulation capability. The influence mechanism of insulation deterioration under long-term repetitive pulses is still unclear. In order to reveal dielectric surface flashover characteristics under RMPs, similar-scale experiments were designed to approximatively resemble the working conditions of a one-stage insulator. Tests on SIL and the aging process were conducted and efforts are made in this paper to determine which condition is more suitable for reducing the occurrence of surface flashovers, preferring a longer SIL and exhibiting the least amount of damage when suffering surface flashover sparks. In addition, we discuss the accumulation effect and the impact of UV content on surface flashovers under long-term RMPs in an SF_6_ or air environment.

This remaining sections in this paper are organized as follows: the experimental setup is described in Section 2. In Section 3, the SIL and aging process experiments are presented. In Section 4, underlying mechanisms of the flashover paths and the accumulation effect are discussed. In Section 5, conclusions are presented.

## 2. Materials and Methods

### 2.1. Experimental Setup

A schematic illustration of our experimental setup is shown in Figure 2. In order to closely mimic the conditions of stage insulators in MGS, the test apparatus, consisting of a repetitive pulsed generator, experimental chamber, measurement instruments, and a PC operating platform, had to be carefully designed to cover the working range of insulators, including voltage and pressure (See Appendix A). RMPs were achieved via a transformer-type pulse generator with an output voltage in the range of 0–150 kV (above the working voltage of one stage in the MGS), max energy supply of ~22.5 J, and a rise time of ~30 μs with variations in PRF from 1 to 50 Hz. Generally, the trigger system controlled the working sequence of charging, discharging, recording the signals, and opening the camera to capture flashover images.

The test chamber was constructed using stainless steel that can sustain pressures from 0.1 Pa to 0.6 MPa. In order to conduct experiments using different gases, a fill port, release valve, and an exhaust flange were attached to the test chamber. In our investigations, SF_6_ with a purity greater than 99.99% was utilized, and compared with compressed dry air. Before each experiment, the chamber was first evacuated to 0.1 Pa using the mechanical pump, filled with SF_6_ and then expelled more than three times. Finally, the chamber was filled with fresh SF_6_ to reach the desired pressure value, making sure that most of the water vapor and air were removed. Thus, the purity of the SF_6_ used during the experiments could be guaranteed [15].

### 2.2. Test Electrodes and Insulator Samples

With the requirement of simulating the quasi-uniform electric field of one rimfire stage in our experiments, a couple of parallel-plane electrodes and insulator samples were designed with diameters of 75 and 40 mm, respectively. In addition, 6 Nylon rods were fixed in a circle through the upper and lower support plates to maintain the balance of the test electrodes and insulator samples and to keep the stress constant from test to test. Meanwhile the upper and lower support plates could be adjusted to fit different sample thicknesses. In order to avoid metal oxidation in the electronegative gas, test electrodes were made of stainless steel (S. steel).

Insulator samples are manufactured from Nylon, polymethyl methacrylate (PMMA), and poly tetra fluoroethylene (PTFE), with a diameter of 40 mm, and thicknesses of 10, 20, and 30 mm, respectively. In addition, the main physical properties of Nylon, PMMA, and PTFE are compared and shown in Table 1 [11,12,16].

The surface condition of the insulators and electrodes directly impact the flashover characteristics, due to their surface roughness, surface micro protrusions, surface defects, dust, moisture, etc. In addition, inevitable microscopic erosion caused by flashover tests easily result in surface deterioration when a sample becomes conditioned, especially under long-term pulses. As such, the test electrodes and insulator samples needed be replaced after ten flashovers to reduce the impact of surface damage. In order to ensure consistency of the experimental conditions, a strict preparation and replace procedure for the test electrodes and insulator samples was put forward. Before a new test, all insulator samples and electrodes were polished using new sandpaper (1200 grit), cleaned with ethanol alcohol, dried at room temperature for 24 h, and finally installed into the test chamber without surface contamination. After ten shots, the old sample was replaced and the electrodes were polished and cleaned again.

### 2.3. Experimental Measurements and Procedures

The test chamber was equipped with a measuring flange to allow desired diagnostics, including voltage, current, and images of flashovers. The applied voltage and flashover voltage were measured by a capacitor voltage divider with a ratio of about 15,000, which was attached to the high voltage output end of the pulse generator. The flashover current was measured using a Pearson 4191 Rogowski coil (Pearson Electronics, Palo Alto, CA, USA) with a sensitivity of 0.01. A protection resistor connected in series with the grounded electrodes was used to reduce the current damage during the flashover tests. When flashover occurred, the repetitive pulse generator was immediately terminated once the flashover current was detected and the Tek TDS3054B oscilloscope (Tektronix UK Ltd., Bracknell, Berkshire, UK) was triggered to simultaneously record both the voltage and current signals (see Figure 3a); a high-speed MotionPro HS-4 (DEL imaging, Woodsville, NH, USA) camera was employed to capture images of the flashover events.

In addition to surface flashover voltage, the surface flashover time delay (*t*_delay_) is also revealing for the surface insulation capabilities of different materials, which is regarded as a crucial parameter in surface insulation design. If a material has a higher flashover voltage with a longer *t*_delay_ than another material, its surface insulation strength can be considered superior. According to the flashover process, *t*_delay_ consists of an avalanche formation time, pre-flashover channel formation time, and a final arc time along the surface. In our experiment, *t*_delay_ (shown in Figure 3b) is defined as the time from 10% of the flashover voltage peak value at the rising edge to 90% of the flashover voltage peak value at the falling edge when flashover occurs, demonstrating the effect of the avalanche formation time and the pre-flashover channel formation time.

In this paper, the pulse number of the applied RMPs, before flashover, is defined as the SIL, which reflects the dielectric surface insulation capability under long-term RMPs, and is closely related to the entire process of the surface streamer, from initiation to development, and finally to flashover.

According to the Martin formula [2], since the exponential dependence of SIL on the flashover voltage, which varies according to different gases, pressures, and materials, there is a great deal of difficulty to obtain the relationship between SIL and above-mentioned factors under the same applied voltage. In order to characterize SIL, we defined the ratio of the applied voltage to the flashover voltage as insulator work coefficient *λ*:(1)$$\lambda =U_{applied}/U_{flashover}$$
where *U_applied_* is the applied voltage and *U_flashover_* is the 100% flashover voltage under the condition of a single pulse.

## 3. Results

### 3.1. SIL under RMPs

The dependences of SIL on gas species, gas pressure, insulator materials, insulator thickness, and PRF were investigated with an insulator work coefficient (*λ*) of 0.9. Since it was found that SILs have a high degree of dispersion, the Weibull distribution [17] of SILs among 10 measurements under the same conditions was utilized to determining the statistics, analyses, and prediction of long-term surface insulation failures. The expression is:(2)$$P\left(N\right)=1-\exp\left[-{\left(N/\alpha_{N}\right)}^{\beta_{N}}\right]$$
where independent variable *N* represents SIL. *P*(*N*) is the cumulative probability of surface insulation failure, i.e., flashover occurs. *α*_N_ is the scale parameter and equals the SIL under a 63.2% flashover probability. *β*_N_ is the shape parameter [18] and reflects the distribution range of the SILs: a larger *β*_N_ indicates a smaller dispersion of SILs. At the same time, the mathematical expectation of *N* is defined as the average SIL:(3)$$\bar{N}=\alpha_{N}\Gamma \left(1+1/\beta_{N}\right)$$
where Γ is the Euler integral function of the second kind.

#### 3.1.1. The Influence of Gas Species

Firstly, SILs of Nylon insulator samples with a 20-mm thickness were tested in air and SF_6_, respectively, at 0.1 MPa and the average *U_flashover_* for each condition were 44.14 kV and 102.08 kV. The Weibull distributions of SILs are shown in Figure 4a and the average SILs in SF_6_ and air were 7534.02 and 758.176 with *β*_N_ of 4.811 and 2.009, respectively, indicating that SILs with less dispersion in SF_6_ are an order of magnitude higher than those in air. The longer SIL in SF_6_ mainly results from the higher insulation strength of SF_6_. Firstly, the strong electronegativity will accelerate electron adsorption and hinder the formation and development of surface streamers due to a lack of carriers. Meanwhile the large molecular diameter of SF_6_ will shorten the mean free path of electrons, which makes it difficult to accumulate energy and produce high-energy electrons for ionization because the ionization potential of SF_6_ is high [19]. In addition, when SF_6_ meets electrons, the polarization will also increase the energy loss, thereby inhibiting the impact ionization process resulting in less seed electrons. Less dispersion of SILs in SF_6_ is likely due to the greater impact of the insulator surface on steamer propagation, mainly resulting in streamer paths along the surface, regarded as the constrained streamer. On the contrary, in air, impact ionization in the gas part and interactions with the insulator surface contribute to more, different streamer paths, thus increasing dispersion.

#### 3.1.2. The Influence of Gas Pressure

SILs of Nylon insulator samples with a 20-mm thickness were tested in air at 0.1 MPa, 0.2 MPa, and 0.3 MPa, respectively, and the average *U_flashover_* for each condition was 44.14, 57.36 and 71.98 kV. The Weibull distributions of SILs are shown in Figure 4b and the average SILs were 758.176, 4465.79 and 20,288, respectively, indicating that the SIL becomes longer at a higher pressure, which has an obvious inhibitory effect on streamer propagation because of the greater energy loss. Meanwhile, the source of charge carriers from the previous pulses was reduced since detrapping and photoemission process become weaker at higher pressures. Among the test conditions, *β*_N_ of 2.009, 3.36 and 3.882 indicate that with the increasing pressure, the dispersion of SILs continuously decreases because streamers are more likely to develop along the surface at higher pressures.

#### 3.1.3. The Influence of Insulator Materials

The dependence of SILs on insulator materials was determined using Nylon, PMMA and PTFE insulator samples with a 30-mm thickness, in air, at 0.1 MPa, and the average *U_flashover_* for each condition was 50.85, 58.46 and 59.86 kV. The Weibull distributions of SILs are shown in Figure 5a and the average SILs were 7032.35, 5075.23 and 3526.39, respectively, indicating that material with a greater dielectric constant (see Table 1) tends toward a longer SIL. The positive correlation between dielectric constant and polarization is attributed to more electrons accumulating along the surface due to the attachment process and the forming of an electric field, which weakens the applied electric field. At the same time, when a streamer has developed along the surface, the material with the greatest dielectric constant is more likely to cause charge carrier adhesion, which also hinders the development of streamers; as such the SIL is improved.

#### 3.1.4. The Influence of Insulator Thickness

SILs of Nylon insulator samples with thicknesses of 10, 20 and 30 mm were tested in air at 0.1 MPa, and the average *U_flashover_* for each condition were 24.94 kV, 44.14 kV, and 50.85 kV. The Weibull distributions of SILs are shown in Figure 5b and the average SILs were 66.3781, 758.176, and 7032.35 with *β*_N_ of 1.964, 2.009 and 2.657, respectively, indicating that as the thickness increased, the SIL increased greatly while the dispersion was reduced. The reason for this is that, when the thickness is small, the electrons induced via field emission near the triple junction have a more significant impact on the local electrical field; as the sample thickness increases, the surface path of the streamer has a greater impact on the SIL since the streamer along surface could be regarded as a constrained streamer because of the electric field, compared with the free streamer in the gas part [20].

#### 3.1.5. The Influence of PRF

SILs of Nylon insulator samples with a 20-mm thickness were tested, respectively, in air and in SF_6_ at 0.1 MPa under different PRFs. The curve of average SILs varying with PRF is shown in Figure 6a. In SF_6_, SIL decreased with increasing PRF since the reduction of the pulse interval is attributed to sufficient energy accumulation, which is required by previous pulses. On the contrary, the variation of SIL dependent on PRF is slighter in air, and sometimes there is even a longer SIL under greater PRF, indicating that the accumulation effect would be weaken in air and details of the causes are discussed in Section 4.2.

It was found that the accumulation effect due to PRF is more obvious on the SIL recovery capability. If flashover occurs when the number of applied pulses is *N*, another *N* pulses are continuously applied to the insulator sample without stopping the pulse generator and the number of flashovers in another *N* pulses is recorded as *N′*. The SIL recovery percent is defined as 1-*N′*/*N* in this paper and the variation with PRF is shown in Figure 6b. The great drop in the SIL recovery percent with increasing PRF indicates that more seed electrons are accumulated to maintain successive flashovers. In addition, as the source of seed electrons, ions, and metastables [21] that survived from previous pulses would be concentrated due to a more frequent ionization with increasing PRF, the adequate energy accumulation can lead to a stable streamer propagation [22].

#### 3.1.6. The Influence of Insulator Work Coefficient

SILs of Nylon insulator samples with a 20-mm thickness were tested in air at 0.25 MPa at 50 Hz RMPs when *λ* was 0.95, 0.925, 0.9, 0.875, and 0.85, respectively, and the Weibull distributions of SILs were obtained (see Figure 7a).

For different *λ*, the average SILs were 692.773, 3294.18, 10,941.7, 30,322.9, and 80,514.8, with *β*_N_ of 3.189, 3.377, 3.692, 3.702, and 10.19, respectively. The variation in SILs, could indicate that the probability of insulator flashover was greatly reduced when working under a lower *λ*; at the same time, the dispersion of SIL was also less than the working condition of *λ*, greater than 0.9. In other words, the predicted SIL was closer to the actual situation when the applied voltage was low. According to the above-mentioned average SILs under different *λ*, the tendency of the average SILs varying with *λ* is shown in Figure 7b, and linear fitting was performed to obtain the empirical formula of the average SILs:(4)$$\mathrm{lg}\bar{N}=A\lambda +B$$

Slope *A* represents the sensitivity of the average SIL to the variation of *λ*. The smaller the absolute value of *A* is, the smoother the SIL curve is, and the wider the work coefficient range that can be selected within the allowable SIL. According to Equation (4), *λ* can be used as a parameter to describe the characteristics of the SIL under RMPs.

### 3.2. Surface Aging Process under RMPs

#### 3.2.1. Successive Flashovers under 1 Hz RMPs

Successive flashovers of insulator samples with a 10-mm thickness were investigated to reveal the difference in the surface aging process between the samples in air and SF_6_. Firstly, in air, at 0.25 MPa, 1 Hz RMPs were applied to Nylon insulator samples and the flashover voltages of 1000 successive shots were recorded (see Figure 8a).

The first flashover voltage was 53.94 kV and after 42 shots the flashover voltage reached its maximum value of 64.38 kV. Among the first 100 shots, an obvious conditioning effect was demonstrated by the phenomenon that the first flashover voltage was less than 95% of the subsequent flashovers. Between 100 and 150 shots, the flashover voltage decreased quickly, and then increased again between 150 and 200 shots. Between 200 and 300 shots, the flashover voltage increased again, and after 300 shots, it decreased again; the initial decrease was rapid, and then the trend is slowed down. Between 300 and 500 shots, this fluctuation repeated itself twice. After more than 500 shots, the flashover voltage showed a trend of picking up again for a few shots and then after 600 shots, the flashover voltage decreased slowly with the fluctuations. Between 600 and 1000 shots, the flashover voltage remained at about 48 kV with a minor drop. In general, the surface insulation performance gradually decreased after 1000 flashovers with the flashover voltage dropping from 47.53 to 44.45 kV in stages. The aging process of the insulators in air presented characteristics of alternating aging and conditioning in the early stage, stable in the middle stage, and the gradual deterioration in the later stage.

In order to conduct the tests in SF_6_ under the similar applied voltages within air, 1 Hz RMPs were applied to Nylon, PMMA, and PTFE insulator samples at 0.1 MPa. Two samples were selected for each material and flashover voltages for each shot were successively recorded, shown in Figure 8b. For PTFE, Nylon, and PMMA insulators, the number of flashovers before surface insulation failure was about 30, 20 and 15 times respectively. Completely different from air, it was found that surface insulation failure bursts in SF_6_ and the insulator surface is irreversibly deteriorated after only a very few shots without a gradual reduction process of flashover voltage such as that in air.

This phenomenon can be explained according to two aspects. It is known that SF_6_ is very sensitive to uniformity of an electric field, in spite of its high hold-off field strength. Once the streamer along the surface develops to a certain extent, the overall electric field distribution is distorted, resulting in a significant decrease in the hold-off field strength. At the same time, in spite of the strong adsorption capacity for low-energy electrons in SF_6_, it drops sharply for high-energy electrons. Generally, the probability of electrons with energy exceeding 1 eV is very low in a relatively uniform electric field [19]. However, once the local electric field concentration occurs in the insulation system, a large number of relatively high-energy electrons will be excited, and more impact ionization between high-energy electrons and SF_6_ molecules is induced, resulting in a decrease in insulation performance [19]. The difference in the flashover paths between air and SF_6_ is also attributed to this phenomenon. More flashovers develop along the surface in SF_6_ resulting in more microscopic erosion across the material surface, and hence a greater decrease in surface insulation. In addition, with successive flashovers, the increased roughness of the surface in SF_6_ induces more electrons accumulated along the surface, and the electric field is distorted, thereby destroying the dielectric/ SF_6_ composite insulation.

Another trend, the dispersion of flashover voltage, is larger in air than in SF_6_. Due to the great difference in flashover voltage between air and SF_6_ at the same pressure, the coefficient of variation, *C*_υ_ (the ratio of standard deviation and average), is utilized to accurately describe the dispersion. Taking Nylon insulator samples with a 10-mm thickness as the example, the PRF-dependent tendency of *C*_υ_ in air and in SF_6_ at different pressures is shown in Figure 9; as are the average flashover voltages of each condition. When the gas pressure increased, the dispersion of every condition decreased, though to different degrees. With PRF increasing from 1 to 50 Hz, *C*_υ_ varied from 0.138 to 0.1 in air and from 0.056 to 0.048 in SF_6_ at 0.1 MPa, and from 0.103 to 0.087 in air and from 0.018 to 0.014 in SF_6_ at 0.2 MPa, respectively.

#### 3.2.2. The Effect of PRF on Surface Aging Process

PRF also has a significant effect on the surface aging process under successive flashovers. When 1, 10 and 50 Hz RMPs were applied to Nylon insulator samples in air at 0.1 MPa, photographs of the surface state after 5000 flashovers were taken. Six photos were taken for each sample and were combined to present the whole circle of the sample surface, shown in Figure 10a; from top to bottom, 1 10 and 50 Hz in turn. It was found that the position of 5000 flashovers at 1 Hz was relatively random, almost all over the whole circle of the insulator surface. Then, the flashover position at 10 Hz was more concentrated than 1 Hz, and the damage to the surface was more serious. Meanwhile the flashover position at 50 Hz was mostly concentrated in only two places and the continued sparks at a fixed location caused ablation of the surface. This indicated that surface flashover occurred at multiple locations in the initial stage. When one of the channels was more severely burned, the surface flashover would frequently occur at that location, especially at a higher PRF. This is the reason why efforts were made to reduce the occurrence of flashover traces and the surface damage caused by flashovers, since once damage formed, the flashover would basically develop along that channel, and a higher PRF intensified this process. At the same time, flashover traces generally developed from the ground side to the high-voltage side due to the movement direction of the electrons; as a result, the ablation traces around the grounded electrode were also more obvious.

Meanwhile, the severity of surface damage increased with increasing PRF (see Figure 10b). According to the flashover traces, the surface damage area could be divided into the electrode contact area and the flashover channel area between electrodes. At 1 Hz, small electrical branches appeared in the electrode contact area, carbonized during flashovers, and gasification channels appeared in the flashover channel area. At 10 Hz, the carbonization of the electrode contact area was intensified significantly, and the growing electrical branches split the surface into small units so that the flashover channel area became rough; the color of the surface was changed after successive flashovers as well, due to the thermal accumulation effect. At 50 Hz, more and more small electrical tree branches continued to develop. The gasification channels spread all over the flashover channel area, and the ablated area on the surface was deeper and longer than that at 10 Hz. Furthermore, the discoloration of the material was intensified, accompanied by the phenomena of wrinkles and ravines that released crystals; obvious melting marks and holes appeared on the surface in the most severely ablated area. At the same time, due to the thermal accumulation effect, the material melted and the gasification jet sputtered to the gas part (see Figure 11).

Flashover current plays a crucial role in surface deterioration during the aging process and the waveforms of its initial stage, 500, 1000, 2000, and 5000 flashovers under 1, 10 and 50 Hz were recorded, respectively (see Figure 12). It can be seen that before there is obvious damage on the surface, the flashover current varies slightly with PRF as the initial state, shown in Figure 12. Then, though the flashover current increases with surface deterioration, getting worse at all PRFs, the increase in flashover current is higher and more rapid at higher PRFs. After 5000 flashovers, the peak flashover current reached 197, 127 and 108 A, respectively, at 1, 10 and 50 Hz, as well as a faster increase in time at higher PRFs. The experimental results show that the discharge positions became more concentrated and the surface deterioration was more serious at higher PRFs. Meanwhile the concentration of flashover positions at various PRFs can be demonstrated by the difference in flashover current, because even after 2000 flashovers, the flashover current at 1 Hz was still lower than that at 50 Hz after 500 flashovers, which caused less damage to the surface. Since a larger flashover current leads to more serious surface damage, and more damage will promote the increase in flashover voltage in turn, the surface deterioration becomes worse at higher PRFs due to the accumulation effect, which can be characterized by the variation in flashover currents during the aging process. It was demonstrated that the difference in flashover currents at various PRFs can reveal the essence of the differences among the surface states, in addition to the correlation between flashover current, the distribution of flashover positions and the degree of surface deterioration are of great importance in the understanding of accumulation effect due to PRF.

#### 3.2.3. Surface Aging Process under 50 Hz

In order to further reveal the deterioration process of insulator surfaces during continuous flashovers, 50Hz RMPs were applied to Nylon insulator samples and the surface states of electrode contact areas after 50, 100, 200, 500, 1000, and 5000 flashovers are shown in Figure 13. The electrode contacted areas were always the initial position of the flashover [23]. When positive RMPs were applied, the electrons from the triple junction were induced via the field emission in the initial stage, and then the electron avalanche was formed. Driven by the local electric field, an accelerated trajectory of electrons spread from the ground side to the high-voltage side. As a result, carbonization traces appeared near the grounded electrode after about 50 flashovers. During continuous flashovers, the carbonization traces gradually became obvious, and tiny electrical branches appeared after 200 flashovers. The material was dented and divided by gasification channels at 500 times, and the carbonization traces tended to spread to the flashover channel area. Due to the thermal accumulation effect, material gasification occurred at 1000 times. As the subsequent flashovers continued to erode the material, grooves formed on the surface in the flashover channel area, and the previous damage of the electrode contact area became more obvious. After 5000 times, the damage of the flashover channel area was more severely carbonized and was accompanied by holes and discoloration. Moreover, the carbonization traces in the contacted high-voltage electrode area, which was not previously obvious, also developed in the flashover channel area with tiny electrical branches. Though the area between the electrodes was ablated and melted, the whole surface had been obviously discolored due to heat accumulation. At the same time, obvious melting marks and holes appeared in the more severely damaged flashover channel area.

In addition, the flashover voltage was tested after 500, 1000, 2000 and 5000 shots, and the reduction from the initial state is shown in Table 2.

Despite suffering damage under successive flashovers, the flashover voltage seemed to drop less than was expected. Hence the SIL of the sample after 5000 times and the fresh sample were tested under 50 Hz with a *λ* of 0.9, and the average SIL was 1240.25 and 12,021.08, indicating that the SIL of the sample after 5000 times dropped greatly and the parameter of the SIL is more suitable to characterize surface insulation strength rather than flashover voltage under long-term RMPs, especially when the insulator has already been damaged.

## 4. Discussion

### 4.1. Influence Mechanism of Flashover Paths

In our investigations, flashover paths made the difference in surface insulation strength, SIL, and the surface aging process. The discharge images of the surface flashovers in air and SF_6_ were taken using a high-speed camera (MotionPro HS-4) (DEL imaging, Woodsville, NH, USA) and it was found that the flashover paths generally had four forms, shown in Figure 14: (1) closely attached to the surface, (2) basically along the surface, (3) developed from one end of the insulator to the gas part, and (4) developed completely in the gas. Allen [14] used photomultiplier tubes to study the propagation characteristics of streamers on the surface of different insulation materials, in 1999, and found that the streamers on the surface had two components: the surface component and the air component. Inspired by this, the flashover discharge paths can be also divided into surface path as (1) and (2) forms, and the gas path as (3) and (4) forms.

#### 4.1.1. Flashover Dispersion

In our investigations, flashover dispersions, including SIL dispersion and flashover voltage dispersion, had a relationship with flashover paths to some extent. Flashover voltage dispersion is characterized by the coefficient of variation, *C*_υ_, and SIL dispersion is characterized by the shape parameter, *β*_N_, according to the Weibull distribution of SILs. Under the same conditions, the dispersion in SF_6_ is much smaller than in air. The streamer along the surface could be regarded as the constrained streamer because charge carriers are attracted towards the solid surface due to the surface electric field, contrary to the free streamer in gas part. Given that constrained streamer means less dispersion, there are more “gas” paths in the air resulting in an increase of flashover dispersion. As the pressure increases, more flashover paths are restricted along the surface, so the dispersion decreases accordingly. In addition, it is also related to the multi-channel flashover sparks along the surface, shown in Figure 15.

Multi-channels occur more frequently in air at lower pressures, because of more random occurrences of flashover positions and more “free” spark channels due to enough drift. With increasing pressure, flashovers become more intense with more “constrained” spark channels, and multi-channel formations are rare, though still observed. There are almost no multi-channels found in SF_6_. Meanwhile, less multi-channels are found when the thickness is increased due to energy loss. *t*_delay_ will be reduced due to the generation of multi-channels, so the flashover voltage at that time is lower than the condition without multi-channels, further increasing the dispersion.

#### 4.1.2. Flashover Paths in Different Gases

At the same time, the flashover discharge traces on the surface of the Nylon insulator samples after 100 flashovers were taken in air and SF_6_, respectively (see Figure 16). It could be also verified that the flashover took more “gas” paths in air, since most of the flashover traces were around the electrode contact area, especially on the grounded side. On the contrary, there were more “surface” paths in SF_6_, since the flashovers left a large number of carbonized channels on the surface, which made the insulator deteriorate rapidly.

Given previous investigations [11,12] conducted by John et al., compared with air, more UV content was detected in the SF_6_ discharge, which was proved by the measurement of optical emission spectra, and would make the flashover paths be more closely attached to the surface. Meanwhile, the presence of the dielectric surface in SF_6_ played a more dominate role in the flashover process because of the photoemission from the surface. In our experiments, it could be assumed that the UV content was gathered from the previous discharge under RMPs. When absent of UV content, the average electron energy was generally estimated to be ~4.5 eV at the flashover threshold. Though there was a considerable improvement up to ~15 eV according to the electron energy distribution function [24], it was still obviously about half that of the first cross-over point (on the order of 30 eV for most dielectrics) for the secondary electron emission (SEE), so that the electrons would be trapped in the bulk of the dielectric when impacting the surface [25]. As a result, the surface would be negatively charged and due to the local electric field, the head of any electron avalanche would be repelled from the surface. Experimental results showrf that the situation was different since more UV content was produced in SF_6_ [26]. Charge carriers were expected to be efficiently stimulated via photoemission because quantum yields for photoemission from dielectric surfaces increase distinctly and the energy of electrons in photon-assisted detrapping is higher than the maximum trap energy [27]. Meanwhile as a strongly electronegative gas, SF_6_ would assist this process more than air. Hence, it was speculated that more electrons that were detrapped from the dielectric were released by the higher UV content gathered in the SF_6_ discharge, forming a net positive charge that attracted the electron avalanche head and let the flashover develop along the surface, and, accordingly, it was easier to form flashover channels and cause more serious damage and ablation on the surface.

In order to further clarify the correlation between the flashover path and UV content, varied by SF_6_ content, pure nitrogen was utilized to form a SF_6_/N_2_ mixture with different SF_6_ contents, because the optical radiation from the nitrogen discharge had almost no effect on the photoemissions [28]. Starting with pure SF_6_, the nitrogen was slowly filled into the test chamber. Figure 17 shows that the percent of “surface” path varied from 52 to 16% with SF_6_ contents varying from 100 to 0%, indicating that more SF_6_ content leads to more surface paths. Furthermore, the percent of “surface” path in N_2_ is less than that in air, again clarifying the correlation between “surface” path and UV content.

#### 4.1.3. Flashover Paths along Different Materials

Flashover paths along different materials are related to the photoemission coefficient which is dependent on the trap level. Charge carriers in shallow paths are expected to be more easily detrapped than those in deep paths [29]. Due to the impact of the binding energy of electrons along the surface, the material of deeper trap levels is expected to have more charge carriers trapped in the trap centers and less detrapped by photoemissions due to the UV radiation. Because PTFE has deeper trap levels and a lower photoemission coefficient [30] than Nylon and PMMA, the UV radiation generated by the previous pulses caused a lower intensity of electron emission on the surface, which caused the flashover path to be repelled from the surface and prefer the more “gas” path. Since the dielectric surface has a greater impact on the flashover path in SF_6_ and the “surface” path will be more likely to cause damage and ablation of the surface, the difference in the flashover paths among the different materials will directly affect the surface aging process of the insulator; if flashovers cannot be avoided, the insulation material with less “surface” path should be chosen to avoid the formation of flashover channels and material ablation caused by long-term operation. The percent of the “surface” path along different materials in 100 flashovers in air and SF_6_ were, respectively, counted, as shown in Table 3.

In order to further reveal the impact of flashover paths along different materials on the damage process of the material surface under RMPs, insulators made of Nylon, PMMA and PTFE were, respectively, exposed to 5000 successive flashovers at 50 Hz, and the comparison of their surface conditions is shown in Figure 18.

The discharge traces on the surface of the PTFE insulators were more widely distributed, and the ablation was slight with no obvious flashover channel area formed because of the smallest “surface” path percent. The discharge traces of Nylon insulators were more concentrated. In addition to the flashover channel area where the melting phenomenon occurred, there were also gasification channels that were not totally penetrated along the surface. The discharge traces of the PMMA insulators were completely concentrated in one place, and the continuous gasification and expansion caused the surface to form huge grooves. At the same time, the flashover channel area near the grounded electrode was cracked, caused by electricity, heat, and force due to the accumulation effect. Though the “surface” path percent of PMMA and Nylon were similar, the degree of surface damage was obviously different due to the heat accumulation. This hypothesis was supported by the glass transition temperature; those of PTFE, Nylon, and PMMA are respectively 327, 220, and 160 °C (see Table 1). The lower glass transition temperature made heat accumulation during the surface aging process easier to cause ablation and melting of the dielectric surface, and once the surface is deteriorated, it causes the flashover position to become more and more concentrated.

### 4.2. Influence Mechanism of PRF

#### 4.2.1. E/p Verse pt_delay_ under Different PRFs

According to our experiments, the PRF affects flashover characteristics. In order to reveal this influencing mechanism, flashover tests under different PRFs were carried out on Nylon insulator samples with a 20-mm thickness in the gas pressure range of 0.1–0.5 MPa in air and 0.05–0.2 MPa in SF_6_. Considering the samples were conditioned and aged during flashovers, the average value of 100% of the flashover voltage under a set of 10 pulses was taken as the flashover voltage at the given PRF, and, similarly, *t*_delay_ at the given PRF was calculated from the average value. Given the positive correlation between flashover voltage and the gas pressure, it should be noted that the flashovers were likely to occur on the rising edge of the applied pulse at a lower pressure. As expected, the increased flashover voltage corresponded to an increased *t*_delay_ and the flashovers occurred nearby, even after the peak of the applied pulse at a higher pressure, due to the increased electron energy loss in the impact ionization because of the decrease in the mean free path.

The surface insulation strength was affected by the electric field distribution, which varied with PRF. Since both air and SF_6_ are electronegative gases, ionization and adsorption must be considered when analyzing the effects of gas breakdown on surface flashover caused by impact ionization. The effective ionization coefficient, αe, is defined as the difference between the ionization coefficient, *α*, and the adsorption coefficient, *η*; *α*_e_/*p* can be expressed as a function of *E*/*p*, as shown below:(5)$$\frac{\alpha_{e}}{p}=\frac{\alpha -\eta }{p}=f_{1}(\frac{E}{p})\sim k_{1}{\left[\frac{E}{p}-{\left(\frac{E}{p}\right)}_{0}\right]}^{2}$$
where *α_e_* is the ionization coefficient in mm^−1^; *p* is the gas pressure in MPa; *E* is the applied electric field strength—in this paper it is calculated from the ratio of the applied voltage to the insulator thickness in kV/mm.

From the electron avalanche to breakdown, the streamer velocity (*ν*), in mm/s, can be calculated using the following equation:(6)$$v=f_{2}(\frac{E}{p})\sim k_{2}{\left(\frac{E}{p}\right)}^{1/2}$$

The Raether’s streamer criterion is *α_e_χ* = constant (*χ* is the critical length of the initial avalanche in mm) [31], and the breakdown formation delay, *τ*, can be calculated by *τ* = *χ*/*ν*, so the product of *pτ* is:(7)$$p\tau =\frac{\alpha_{e}\chi }{f_{1}(\frac{E}{p})f_{2}(\frac{E}{p})}=\frac{A}{{\left[\frac{E}{p}-{\left(\frac{E}{p}\right)}_{0}\right]}^{2}{\left(\frac{E}{p}\right)}^{1/2}}$$

Since the initial charge carriers in the gas/solid flashovers are related to the partial discharge at the triple junction, and the main source of the surface charge is from the gas part, according to Equation (7), it can be illustrated that *pt*_delay_ is also the function of *E*/*p* during the flashover process [32]. In order to compare the surface insulation strength at different pressure ranges, the relationship between *E*/*p* and *pt*_delay_ was plotted at different PRFs, in air and SF_6_, respectively, as shown in Figure 19. It can be seen that as the PRF increases, the overall trend develops to the left. In addition, when the air pressure is beyond 0.2 MPa, the *E*/*p* at each PRF decreases sharply, which indicates that the impact ionization in the gas part and space charge dominate the flashover under RMPs in air. Furthermore, the value of (*E*/*p*)_0_ can be calculated from the nonlinear fitting curve and the PRF-dependent tendency of (*E*/*p*)_0_ is shown in Figure 20. For the same *pt*_delay_, the higher the PRF, the smaller the *E*/*p*, and the value of (*E*/*p*)_0_ expands lower. In air, the (*E*/*p*)_0_ of a single pulse is 12.66 and (*E*/*p*)_0_ for PRF it drops from 10.32 to 9.76. In SF_6_, the (*E*/*p*)_0_ of a single pulse is 30.73 and the (*E*/*p*)_0_ of PRF drops from 25.7 to 24.2. Similarly, there is a plateau period of (*E*/*p*)_0_ during the lower PRF in both air and SF_6_, though the change of (*E*/*p*)_0_ in SF_6_ is slightly bigger than that in air. It should be noted that the differences among the conditions of different PRFs is much smaller than the differences between the conditions of a single pulse and any PRFs. It is illustrated that the value of (*E*/*p*)_0_ represents the surface insulation strength under RMPs in a compressed gas environment, based on the influence of the electric field distribution at various PRFs.

#### 4.2.2. Understanding of the Accumulation Effect on Flashover under RMPs

Surface flashover under RMPs is dominated by electric field distributions and the accumulation effect, including the gas part, the solid part and their interactions. According to the avalanche-to-streamer-to-flashover mechanism, a flashover process usually can be divided into three stages and each stage is related to the accumulation effect, including main physical processes (see Figure 21). Firstly, charge carriers are mainly induced via field emissions and field-assisted hot electron emission near electrodes, especially triple junctions. Because of the inevitable imperfect contact between electrodes and insulators, the severely distorted electric field stimulates a partial discharge at the triple junction and the current formed by the charge carriers is promoted due to micro metal protrusions, which are unavoidable on electrodes. The accumulation effect contributes to the difference in the current source between a single pulsed flashover and a flashover under RMPs. Secondly, the streamer is initiated from the electron avalanche when the Raether’s criterion [31] is reached. Along the insulator surface, propagation of the streamer is affected by the accumulation effect agents, including trapping center holes and electrons, positive and negative ions, electrons from ionization collision, and metastable species. Considering the severely distorted local electric field around the triple junction, the distribution of the residual accumulation effect agents plays a dominant role in the initiation of subsequent surface streamer formed by charge carriers. Finally, a flashover occurs when streamer channels remain along the surface.

The influence mechanism of PRFs is always explained by the accumulation effect. Under repetitive pulses, electrons, ions and metastable species will survive from the previous pulses and affect subsequent pulses [17], this is the so called the accumulation effect. According to different decay times, various accumulation effect agents exist in the gas/solid composite insulation and they can be divided into short-term agents and long-term agents. Usually regarded as short-term agents, positive ions, negative ions, and electrons, mainly in the gas part, are induced via diffusion, recombination, and drift. When it comes to long-term agents, metastable species, surface trapped holes and electrons, surface structure deterioration, and heat accumulation [16] have relationships with the surface part. It is known that the lifetime of some metastable species is over one second. Therefore, abundant metastable species will still survive along the dielectric surface, even under the condition of only 1 Hz, which will make the curve of the *E*/*p* vs. *t*_delay_ quite different from the conditions of a single pulse. Because of the accumulation effect, more seed electrons will be triggered to improve the avalanche of electrons and surface streamer. These seed electrons mainly include two parts: the combination of positive ions and the metastable species generated in the previous pulses will induce the secondary electrons, and the other is the gradual ionization of metastable species. It can be assumed that the reduction of (*E*/*p*)_0_ with increasing PRF primarily results from the growth in the number of seed electrons, which come from residual ions and metastable species [18]. Because of sufficient diffusion and drift at lower PRFs, the distribution of residual charge carriers in the gas part is less concentrated than that at higher PRFs, and, consequently, the SIL drops very little and even fluctuates with PRFs, especially at a lower PRF in air because of more “gas” paths.

Different from the gas breakdown, the surface charge phenomenon is a unique dielectric response, and plays a profound role in thoroughly understanding the flashover process [33]. Investigations in this paper speculated that both the surface insulation capability, the SIL, and the surface aging process are sensitive to surface charge accumulations [34]. Due to the difference in flashover paths, the influence mechanism of the surface charge must take the difference in the accumulation effect agents into account. In the weakly electronegative air, the accumulation and attenuation of surface charges mainly comes from the impact ionization in the gas part. At that time, the electric field formed by the surface charges is comparable to the applied electric field, and the distorted electric field distribution will inhibit the surface streamer. In the strongly electronegative SF_6_, flashover paths are more likely along the surface, so the photoemissions from the surface should be considered. In addition, the minimum deposited surface charges induced by the combined effect of trapping and detrapping process [35] have a profound impact on the flashover process. In a previous study [36], the trap centers were assumed to be evenly distributed, and traps were divided into two types: shallow and deep. Considering the interaction layer between the dielectric and the surface streamers due to the impact of the surface charge binding energy on electrons, the material with more, deeper traps tended to have more trapped charge carriers and less were available for impact ionization with the surrounding gas environment. With the presence of more UV content in SF_6_ flashovers, the process of the seed electrons, gathering from photoemissions, was assisted.

Under RMPs, the SIL reflects the time for the streamer to penetrate the entire surface and establish a flashover, and the seed electrons survived from previous pulses due to the accumulation effect and will promote the development of the surface streamer. This tendency will be inhibited at a high pressure, so the SIL becomes longer when the pressure increases. For different gases, the streamer speed varies with different streamer paths, whether they are surface or gas. For different materials, the charge adsorption capacity of the surface plays a key role in streamer propagation. The charge accumulated along the surface will distort the background electric field and forbid streamer development, which, in turn, affects the propagation speed of the surface streamer, and the material with more charge attachment leads to a slower streamer speed and a longer SIL.

The surface deterioration and the heat accumulation under long-term RMPs mainly affect the aging process during continuous flashovers. Under RMPs, it is obvious that the heat loss during the pulse interval will decrease with the increasing PRF, in other words, the higher PRF will promote heat accumulation. Our experimental results also confirmed that the aging process of the insulator surface under continuous flashovers is accompanied by thermal aging, and the material deterioration caused by the flashover channels causes not only electrical tree branches and carbonized deposition, but also material gasification, ablation, and melting. Overall, our results support the hypothesis of heat accumulation contributing to surface deterioration under long-term RMPs and the increase in PRF aggravating the damage to the insulator surface.

## 5. Conclusions

In the present investigations, flashover experiments were carried out on Nylon, PMMA, and PTFE insulator samples between parallel-plane electrodes in SF_6_ and air at various pressures. The influence of surface path and the accumulation effect on the SIL and the aging process were discussed. The following conclusions can be drawn:

Based on the Weibull distribution, an empirical SIL formula under different insulator work coefficients (*λ*) is proposed. *λ* is a more suitable parameter to characterize the SILs of different materials under RMPs. Under the same *λ*, the SIL in SF_6_ is an order of magnitude longer than in air, and the material with a higher dielectric constant has a longer SIL.

During the continuous flashovers, the aging process in air has the following characteristics: alternate appearance of aging and conditioning in the initial stage, stability with little decline in the middle stage, and gradual deterioration in the later stage. SF_6_ has the characteristics of a sudden insulation failure suffering very few flashovers.

The accumulation effect, due to PRF, has a profound impact on SIL and the aging process. The relationship between *E*/*p* and *pτ*_delay_ shows that the electric field varies with different PRFs and the value of (*E*/*p*)_0_, decreasing with increasing PRFs and less than a single pulse condition can characterize the surface insulation strength under RMPs.

Flashover paths make a difference in the flashover characteristics. Due to the photoemissions induced by the UV content gathered from previous flashovers, more “surface” path in SF_6_ leads to a lower dispersion of flashover and SIL, sudden insulation failure under RMPs, and more damage to the surface during the aging process. The positive correlation between the “surface” path and the SF_6_ content was demonstrated by the percent of “surface” path, varying from 53 to 16% with SF_6_ contents decreasing from 100 to 0%. For different materials, the smaller photoemission coefficient tends to less “surface” paths.

The aging process was promoted by the heat accumulation effect caused by the increasing flashover current, which accelerated the surface degradation and cause the flashover position to be more concentrated at a higher PRF. We should choose the material with fewer “surface” paths and a higher glass transition temperature to avoid the formation of flashover channels and weaken the ablation and deterioration of the dielectric surface.

## Figures and Tables

**Figure 1 materials-14-03343-f001:**
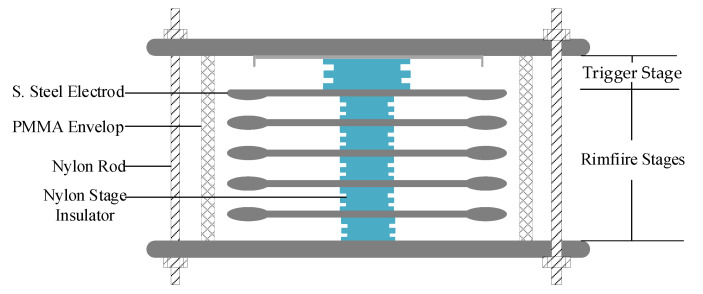
Schematic diagram of MGS.

**Figure 2 materials-14-03343-f002:**
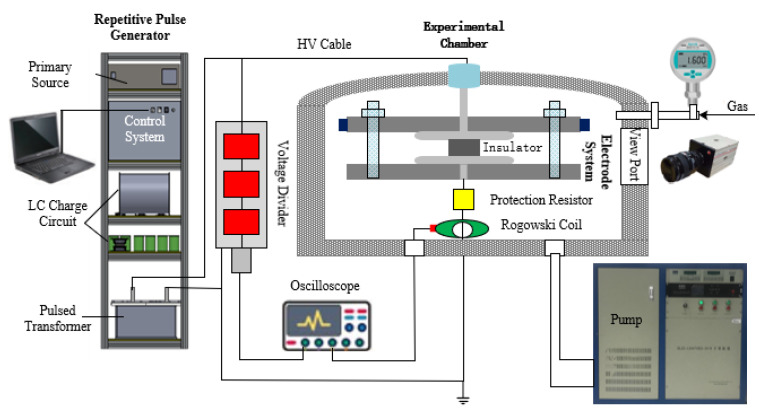
The schematic diagram of the experimental setup.

**Figure 3 materials-14-03343-f003:**
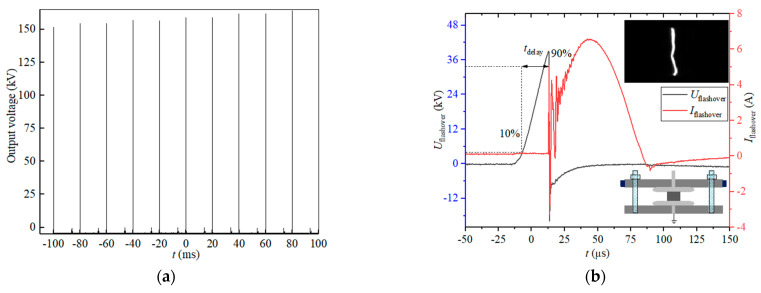
(**a**) Output of the generator and (**b**) flashover waveforms of voltage and current.

**Figure 4 materials-14-03343-f004:**
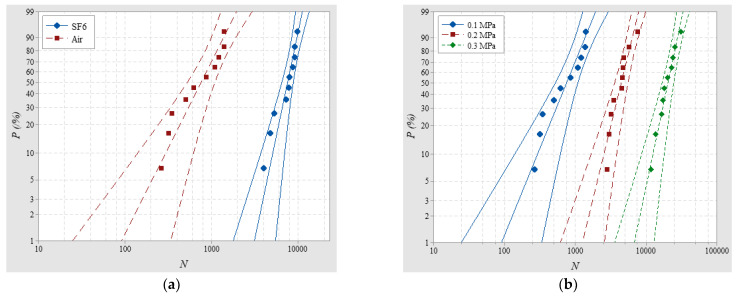
SILs vary with (**a**) different gases and (**b**) different pressures.

**Figure 5 materials-14-03343-f005:**
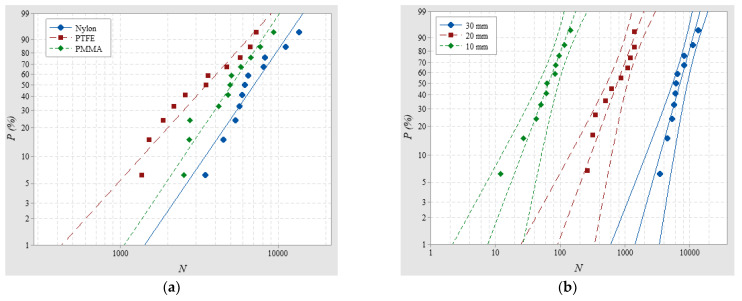
SILs vary with (**a**) different materials and (**b**) different thicknesses.

**Figure 6 materials-14-03343-f006:**
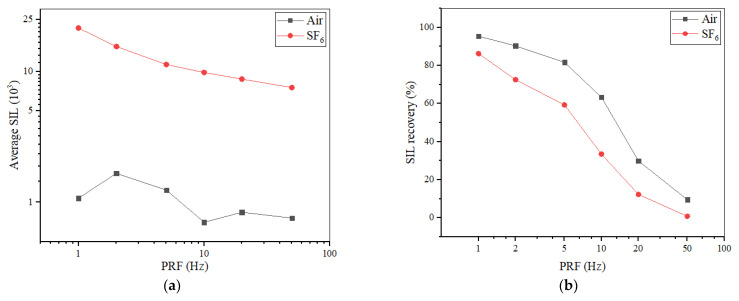
(**a**) SIL and (**b**) SIL recovery percent vary with PRF.

**Figure 7 materials-14-03343-f007:**
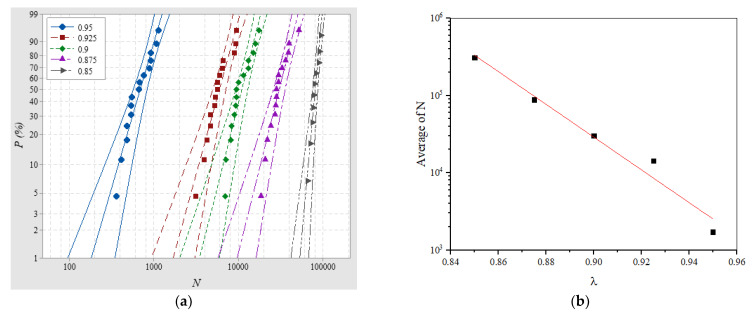
(**a**) SIL distribution and (**b**) average SIL varies with *λ*.

**Figure 8 materials-14-03343-f008:**
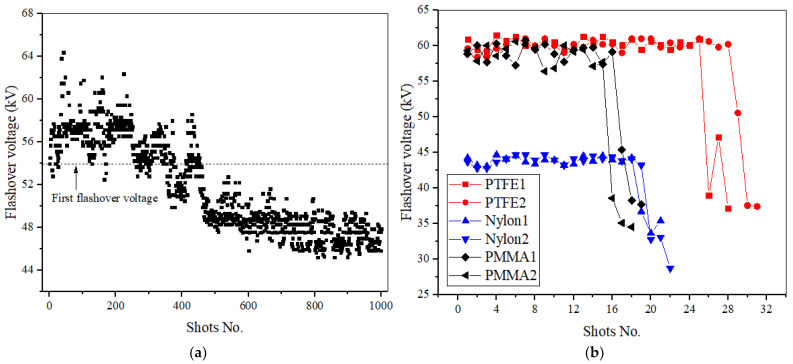
Successive flashover voltage under 1 Hz pulses in (**a**) air and (**b**) SF_6_.

**Figure 9 materials-14-03343-f009:**
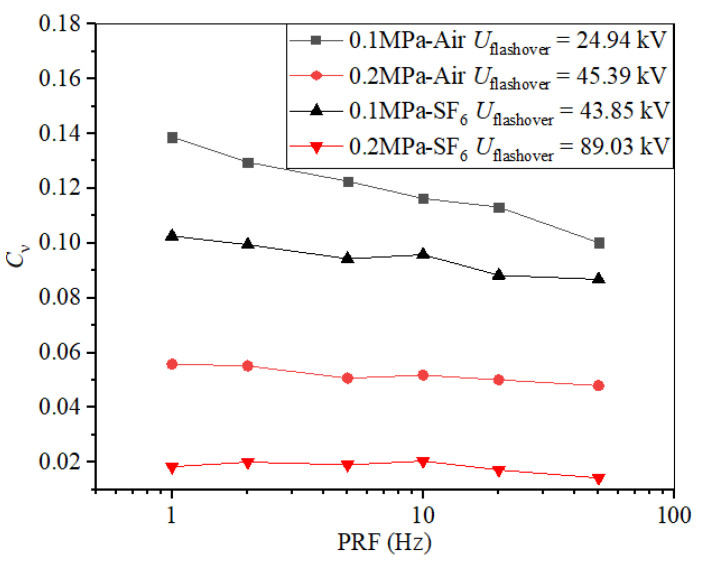
Dispersion of flashover voltage in SF_6_ and air.

**Figure 10 materials-14-03343-f010:**
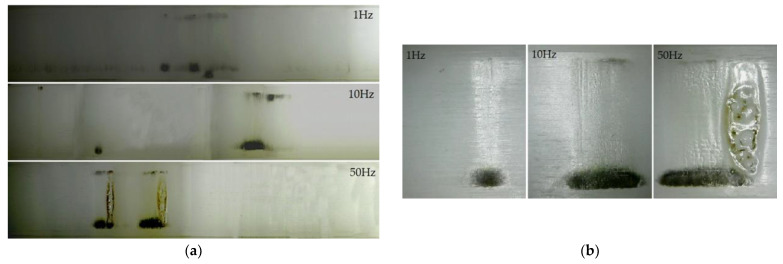
Image of discharge traces including (**a**) the whole surface and (**b**) flashover channels area after 5000 flashovers under different PRFs.

**Figure 11 materials-14-03343-f011:**
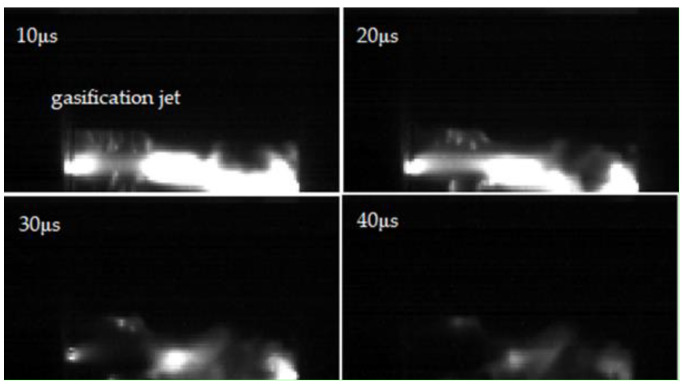
Material gasification during flashover.

**Figure 12 materials-14-03343-f012:**
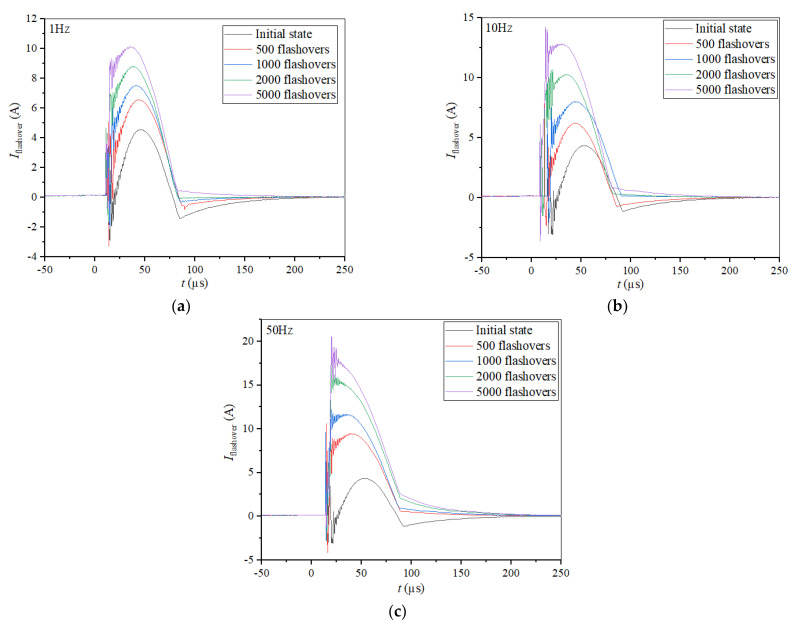
Flashover current during the aging process at (**a**) 1, (**b**) 10 and (**c**) 50 Hz.

**Figure 13 materials-14-03343-f013:**
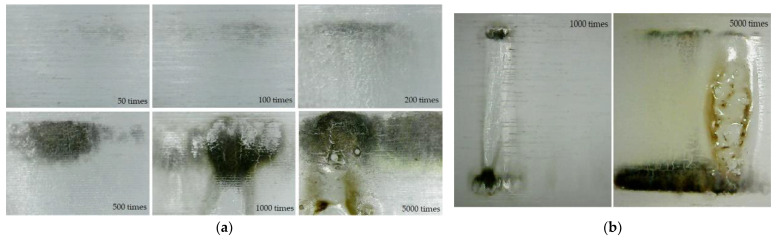
Surface state of (**a**) electrodes contacted area and (**b**) flashover channel area.

**Figure 14 materials-14-03343-f014:**
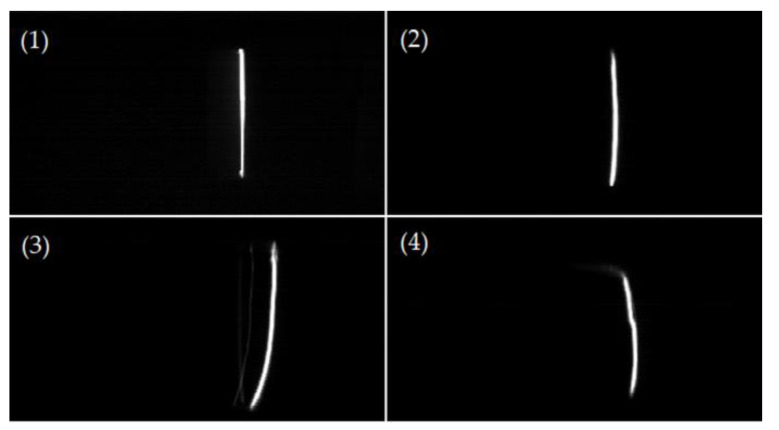
Four kinds of flashover paths.

**Figure 15 materials-14-03343-f015:**
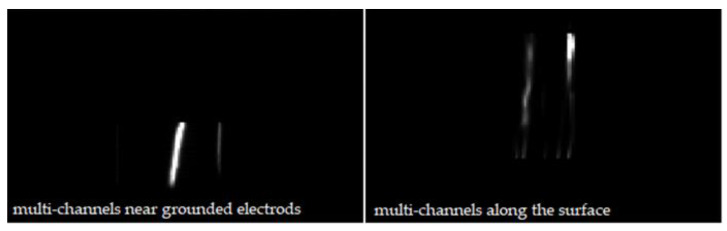
Multi-channel phenomenon.

**Figure 16 materials-14-03343-f016:**
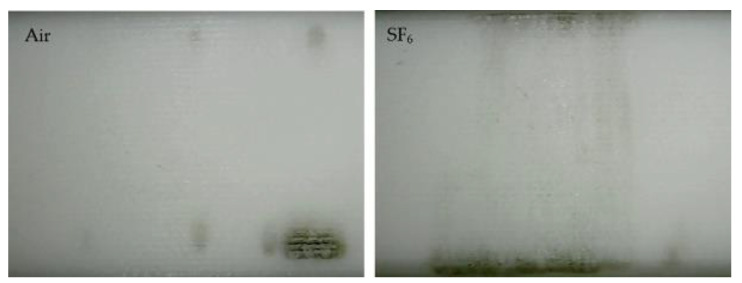
Image of discharge traces after 100 flashovers in air and SF_6_, respectively.

**Figure 17 materials-14-03343-f017:**
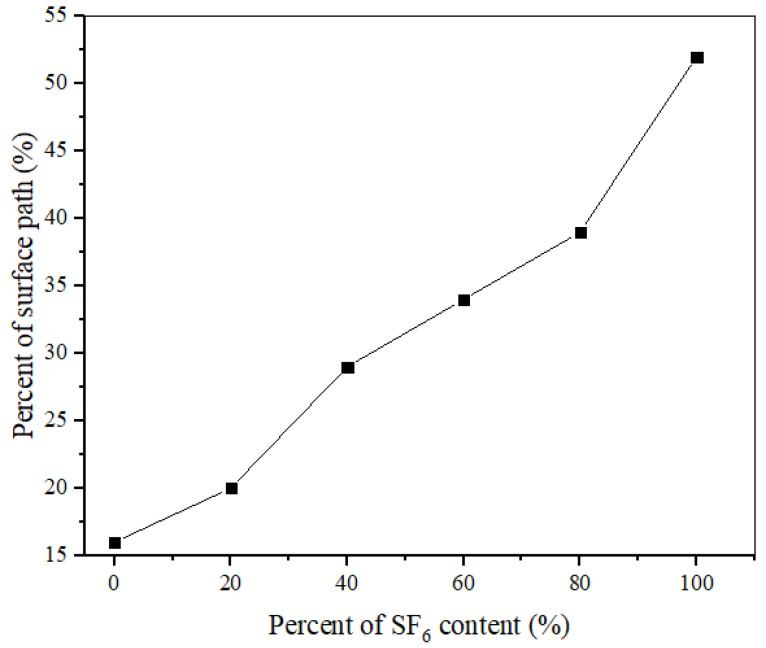
The percent of surface path at the various SF_6_ contents.

**Figure 18 materials-14-03343-f018:**
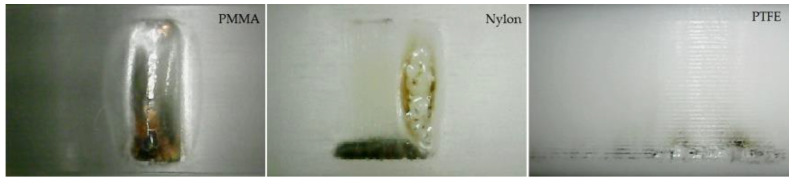
Surface damage of different materials under 50 Hz RMPs.

**Figure 19 materials-14-03343-f019:**
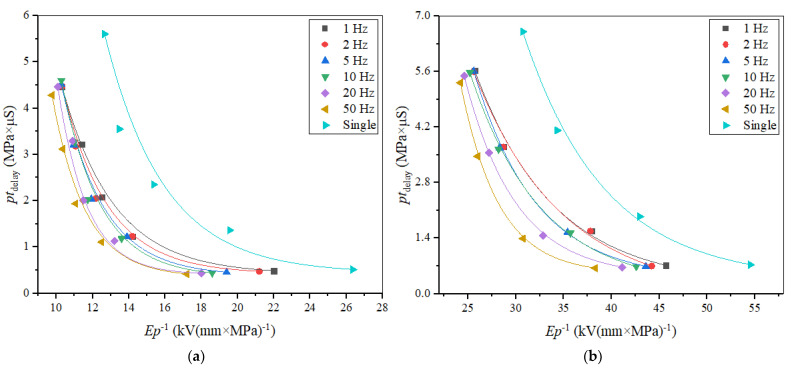
Dependence of *E*/*p* on *pt*_delay_ in (**a**) air and (**b**) SF_6_.

**Figure 20 materials-14-03343-f020:**
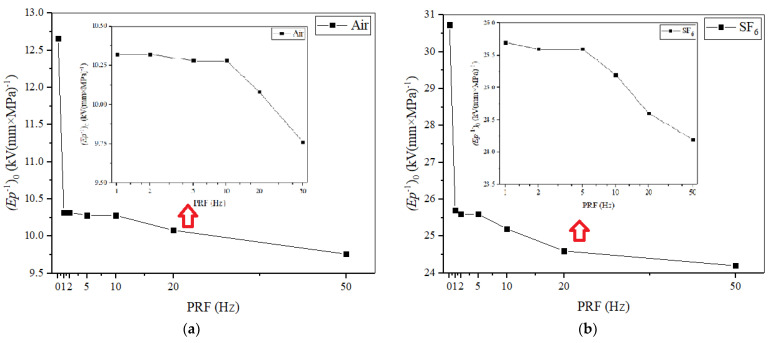
(*E*/*p*)_0_ varies with PRF in (**a**) air and (**b**) SF_6_.

**Figure 21 materials-14-03343-f021:**
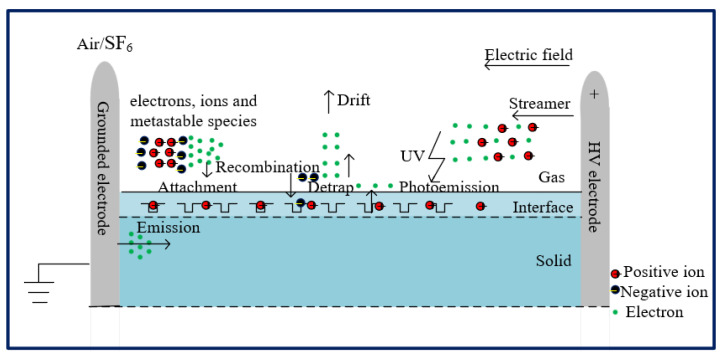
Physical processes involved in flashover under RMPs.

**Table 1 materials-14-03343-t001:** Main physical properties of Nylon, PMMA, and PTFE.

Materials	Nylon	PMMA	PTFE
Dielectric constant	5.0	2.6	2.1
Surface resistivity (Ω/m^2^)	5 × 10^10^	10^14^	10^15^
SEE coefficient	2.42	2.71	2.12
Melting point (°C)	220	150	327
Boiling point (°C)	452	-	400

**Table 2 materials-14-03343-t002:** Reduction of flashover voltage during the aging process.

Shots	500	1000	2000	5000
Reduction	12.04%	14.91%	16.24%	18.67%

**Table 3 materials-14-03343-t003:** Percent of surface path along different materials in air and SF_6_.

Gas	Nylon	PMMA	PTFE
Air	23	20	0
SF_6_	52	63	28

## Data Availability

The data can be obtained from the corresponding author on request.

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
