# Peer review of "Dielectric Surface Flashover under Long-Term Repetitive Microsecond Pulses in Compressed Gas Environment"

_materials, 2021, doi:10.3390/ma14123343_

Round 1

Reviewer 1 Report

This paper describes the surface insulation lifetime by applying repetitive micro-second pulse voltages for three different dielectric materials in SF6 and dry-air gases for evaluation of multi-gap switch in high-power microwave system. Many useful information is included in the paper and is maybe felt interest by researchers and technicians in same research community. The study is based on original work and has enough novelty. Followings are comments for improvement of paper quality.

  1. Page 2, Fig. 1: The text (labels and words) on the figure 1 is too small and hard to identify. Please modify the figure for easily readable.
  2. Page 2, paragraph 1 from the bottom: Please add the information for energy supplied from the high-voltage power supply. The input energy affects lifetime of the materials.
  3. Page 3, paragraph 3: In the abstract and experimental result sections, three materials of Nylon, PMMA and PTFE are appeared. However, the four materials of PI, Nylon, PMMA and PTFE are appeared in this paragraph. Which is correct? Please check it. And please supply the information of acronyms of PI, PMMA and PTFE. And please add information of the constants of physical property of Nylon, PMMA and PTFE, such as dielectric constant, boiling point, and secondary electro emission coefficient.
  4. Page 4, Figure 3(a): Why does the current flow before the breakdown in period from t=-20 to 15 ms shown in Fig. 3(a)? The text (labels and words) on the figure (a) is too small and hard to identify. How much does the discharge current after the breakdown? The discharge current after the breakdown is over-ranged. The value of the discharge current maybe affects the surface insulation lifetime strongly. So, please add the information about the value of current amplitude after the breakdown.
  5. Page 5, paragraphs 1 and 2 from the bottom: Please add the information about values of Uflashover for each condition.
  6. Page 6, paragraphs 1 and 2 from the bottom: Please add the information about values of Uflashover for each condition.
  7. Page 9, paragraph 3: Please add the information about values of Uflashover for each condition.
  8. Page 10, paragraph 1: Is the discharge current at all frequencies the same value? Please add the description of discharge current for various pulse repetition rate. The discharge current and the pulse repetition rate affects carbonization or glassing of each material.
  9. Page 14, Figure 17: How much does the surface path percentage at nitrogen only? If you have data, please add the data on Fig. 17.

End.

Author Response

Dear reviewer,

Yours, Tianyu Lin

Reviewer 2 Report

The authors present in a detail manner the dielectric surface flashover phenomena for high power microwave pulses. It is very comprehensive and if i understood correctly it is the first time the authors describe their technique. Impact wise i believe it could be of interest for Materials community but in this form the paper suffers on several chapters:

 the abstract is not quite what an abstract should be. Just the main points of the paper , the writing with ";" does not help

The introduction has little of the actual literature on the topic. I would like to see where does this system stands with other similar ones. what are the improvements over existing system and what are the goals of the paper. Overall is uneven written.

The quality of the images is low, please add better quality

all over the results part, the presentation in English is quite heavy and difficult to follow. Some statements come out of nowhere: raw 314 regarding the electron energy.

Mu suggestion would be for the authors to think of using some of the info in the paper in supplementary information to help make a leaner paper and more easy to follow for the readers.

Intensive changes are required before publication. I would like to encourage the authors to consider some of my comments and resubmit

Author Response

Dear reviewer,

Yours, Tianyu Lin

Round 2

Reviewer 2 Report

The authors took into consideration my suggestions.